# Effect of Inulin Intervention on Metabolic Control and Methylation of *INS* and *IRS1* Genes in Patients with Type 2 Diabetes Mellitus

**DOI:** 10.3390/nu14235195

**Published:** 2022-12-06

**Authors:** Ollin Celeste Martínez-Ramírez, Azucena Salazar-Piña, Ximena Cerón-Ramírez, Julieta Rubio-Lightbourn, Fernando Torres-Romero, Leonora Casas-Avila, Clementina Castro-Hernández

**Affiliations:** 1Facultad de Nutrición, Universidad Autónoma del Estado de Morelos, Cuernavaca C.P. 62350, Morelos, Mexico; 2Departamento de Medicina Genómica y Toxicología Ambiental, Instituto de Investigaciones Biomédicas, Universidad Nacional Autónoma de México, Ciudad de México C.P. 04510, Mexico; 3Laboratorio de Medicina Genómica, Instituto Nacional de Rehabilitación, Ciudad de Mexico C.P. 14389, Mexico; 4Unidad de Investigación Biomédica en Cáncer, Instituto Nacional de Cancerología, Instituto de Investigaciones Biomédicas, Universidad Nacional Autónoma de México, Ciudad de México C.P. 14080, Mexico

**Keywords:** agave inulin, *INS*, *IRS1*, T2DM, epigenetics

## Abstract

Background and Aims: Currently, treatments are being sought to improve the control of type II diabetes mellitus (T2DM), and inulin has been shown to be effective in reducing glucose levels and other metabolic control parameters. These effects on metabolic control may be associated with changes in the epigenetic modulation of genes of the insulin pathway. Therefore, our objective is to determine the effect of agave inulin in metabolic control parameters and in *INS* and *IRS1* genes’ methylation in T2DM patients. Methods: This was a longitudinal experimental study with 67 Mexican participants who received an intervention of inulin agave (10 g daily) for 2 months. The methylation of the *INS* and *IRS1* genes was determined by MSP. Results: For the *INS* gene, we found a significant decrease in the proportions of T2DM patients with methylated DNA after inulin intervention (*p* = 0.0001). In contrast, the difference in the proportions of the unmethylated *IRS1* gene before and after the inulin intervention was not significant (*p* = 0.79). On the other hand, we observed changes in the number of T2DM patients’ recommended categories for metabolic control depending on the methylation of *INS* and *IRS1* genes before and after treatment with inulin. Conclusion: For the first time, we report the modification in the methylation of two genes, *INS* and *IRS1*, of the insulin pathway and provide information on the possible relevant role of epigenetics as a key factor in positive changes in metabolic control parameters by inulin intake in T2DM patients.

## 1. Introduction

Type 2 diabetes mellitus (T2DM) remains a disease of great importance worldwide due to its high and increasing prevalence. In recent decades, the incidence of diabetes in Mexico has progressively increased from 5.8% in the National Health Survey 2000 [1] to 7.0% in the National Health and Nutrition Survey 2006 [2], to 9.2% in 2012 [3], and to 10.3% in 2018. This latest prevalence represents just over 8.6 million people with a prior medical diagnosis of the disease in Mexico in 2018 [4]. T2DM is a complex disease developed by the progressive loss of insulin secretion in β cells of the pancreas and/or by decreased insulin sensitivity in peripheral target organs that is often associated with age, obesity, lack of physical activity, first-degree family history, and/or with a strong genetic predisposition [5]; however, the genetic etiology of T2DM is still poorly studied.

Epigenetics studies the mechanisms that regulate gene expression without modifying the DNA sequence. These mechanisms include DNA methylation, chromatin modification, and the action of noncoding RNAs and allow genes to be expressed or silenced depending on several conditions, such as biological, lifestyle, and environmental factors, indicating that cells undergo these challenges throughout life [6,7]. DNA methylation is the most studied mechanism and involves the covalent attachment of a methyl group to a cytosine residue to generate 5-methylcytosine (5mC) [7,8]. This union occurs specifically in cytosine-guanine dinucleotides, which are grouped in the genome as repetitive sequences constituting the so-called CpG islands [9].

These CpG islands are especially abundant in gene promoter regions and other regulatory areas. The methylation of CpG islands within promoter regions is associated with gene silencing, and this process is dynamic; unmethylated sequences are capable of being methylated, and methyl groups can be lost [7]. These alterations appear in response to environmental and lifestyle factors, such as diet, exercise, alcohol use, or tobacco use, and they are associated with changes in gene expression and pathological dysfunctions [6]. Several studies have reported the relationship between genetic variation and DNA methylation with respect to T2DM, but it is unknown whether DNA methylation is a mediator in the disease pathway or if it is altered in response to the disease state [10,11,12]. Despite the exponential increase in epigenetic research in the last decade, the status of alterations in DNA methylation in the blood of human subjects with T2DM remains limited [13]. An approach involving candidate genes for quantifying the methylation status of specific CpG sites within genes associated with the development or risk of developing T2DM is increasingly being utilized. Insulin (*INS*) is the main product synthesized by pancreatic *β*-cells [14], and a relationship has been found between the methylation of the *INS* gene promoter and the regulation of *INS* gene expression in both pancreatic islets and *β*-cells [15,16]. Moreover, insulin receptor substrate 1 (*IRS1*), a substrate of the insulin receptor tyrosine kinase, is a critical element in insulin signaling pathways, and mutations in this gene have been reported to play an important role in the determination of susceptibility to T2DM-related traits [17,18]. Due to the increasing prevalence of T2DM, it is important to improve the results of T2DM treatments. The actions used to control this disease include multidisciplinary interventions, pharmacological treatments, and alternative options that are intended to help with the pharmacological treatments of T2DM in addition to exerting an impact on metabolic control and the prevention of T2DM complications. Among the existing alternative options, agave inulin, a highly water-soluble oligosaccharide currently used as a prebiotic [19], has been shown to be effective in reducing plasma glucose levels in diabetic patients. In obesity, dyslipidemia, and mouse model studies [20], the level of GLP-1 in serum has also been shown to be increased, affecting IL-6 secretion, IL-6 production, and hepatic gluconeogenesis, which results in moderate tolerance to insulin and blood glucose lowering [21], thereby promoting insulin secretion, promoting pancreatic β cell proliferation, controlling muscle cell glycogen synthesis, controlling fiber amounts, and providing satiety [22].

For the abovementioned factors, we hypothesized that methylation of the *INS* and *IRS1* genes would change while improving the metabolic control parameters in T2DM patients. 

## 2. Methods and Materials 

### 2.1. Studied Population

An intervention with agave inulin was performed on outpatients of the Hospital of Huitzuco Guerrero, Mexico, who were diagnosed in 2018 with T2DM. The subjects were invited to participate in the present study directly in the outpatient clinic and by telephone. The outpatients who agreed to participate in the protocol signed an informed consent letter.

Prior to this project, an experimental, longitudinal study was performed. The studied population consisted of 67 outpatients with an established diagnosis of T2DM. The inclusion criteria for the selected outpatients were as follows: Age between 30 and 70 years; previous diagnosis of T2DM for at least 12 months; under pharmacological treatment for at least 6 months; fiber intervention of less than 30 g; no gastrointestinal or thyroid diseases; not pregnant or breastfeeding; no intervention with prebiotics, probiotics, lipid-lowering drugs, or antibiotics; and no smokers or alcohol users.

It is important to mention that in this study, the controls were the subjects before the intervention with inulin, that is, each subject was its own control. This design is accepted in longitudinal studies since the biases in the study decrease because the data on age, sex, socioeconomic level, etc., do not change before and after the intervention.

### 2.2. Intervention with Agave Inulin

The intervention was performed for a total period of 8 weeks. At the initial visit of the studied population, a peripheral blood sample was obtained, and the following data were collected: Biochemical data, anthropometric analyses, body composition, and questionnaires about sociodemographic data, adherence to the basic drug for T2DM, food registry, and physical activity.

The participants were given one agave inulin sachet (10 g) every day for 4 weeks. Then, the abovementioned measures were determined, and the participants were given additional inulin sachets for 4 weeks. At the end of the 8-week intervention, a blood sample was collected, and the abovementioned measures were determined.

During the intervention with inulin, we maintained communication by telephone with the outpatients, and in the few cases in which there was difficulty in communicating, we went to the participants’ homes to conduct a timely follow-up.

Inulin is a safe supplement and was approved by the FDA (Food and Drug Administration, FDA) as a safe substance in 2003 [23]. Daily intakes for the U.S. and Europe have been estimated at up to 10 g. On the other hand, some diets can contain considerable amounts of inulin or oligofructose (up to 20 g) [24]. Due to this, the dose that was provided to the participants from start to finish was not intolerable at any time by the patients.

### 2.3. Body Composition

The anthropometric measurements included weight and height for the determination of the body mass index (BMI) using the following formula: BMI = weight (kg)/height (m^2^). The BMI was divided into normal weight (BMI 18.5–24.9 kg/m^2^), overweight (BMI 25–29.9 kg/m^2^), obesity grade I (BMI 30.0–34.9 kg/m^2)^, obesity grade II (BMI 35.0–39.9 kg/m^2^), and obesity grade III (BMI > 40.0 kg/m^2^) according to the classification of the World Health Organization [25]. The waist circumference, fat mass, and muscle mass were obtained with a Tanita HBF-514C scale and a SECA portable stadiometer (Medical Scale Model 214, Seca Corp., ON, Canada).

### 2.4. Dietary Evaluation

The dietary evaluation was performed with 24-h recall during the 8 weeks, and the obtained data were analyzed using the AZ nutrition program (version 23.8), which refers to the System Mexican Food Equivalents 4th edition [26]. The reference indicators used are based on the intake recommendations for nutrients for the Mexican population.

### 2.5. Biochemical Data

The analyzed biochemical data included fasting glucose, triglycerides, total cholesterol, LDL cholesterol (c-LDL), and uric acid. These parameters were determined before and after the intervention with the agave inulin.

### 2.6. DNA Extraction and Methylation-Specific PCR (MSP) Conditions 

Before and after the intervention with agave inulin, 5 mL of peripheral blood was collected from all participants, isolated, and stored at −20 °C. Genomic DNA extraction was performed using the peripheral blood using a standard separation procedure by salinization as previously described [27]. The DNA samples were quantified by UV spectrophotometry using a Nanodrop 1000 (Thermo Fisher, Waltham, MA, USA) to determine the concentration and purity of the DNA. Genomic DNA, 500 ng, was treated with sodium bisulfite with the EZ DNA Methylation kit (ZIMO, 5005), which is the gold standard method for DNA methylation analysis. 

CpGenome universal methylated DNA (Merck Millipore, Burlington, MA, USA) was used as a positive control for methylated DNA and CpGenome universal unmethylated DNA (Merck millipore) was used as a control for unmethylated. Template-free H_2_O was included as a negative control for the MSP.

The CpG sites studied are in a CpG island in the promoter of *INS* and *IRS1* genes. The primers used were designed by the Methyl Primer Express v.1.0 software (Applied Biosystems, Waltham, MA, USA) from data deposited in GenBank. 

For the *INS* gene, the region −1500 to +100 was included (chr11:2155610-2155813). Primers sequences for the methylated reaction were 5′-TATAGTACGGTTGGGTCGC-3′ (forward primer) and 5′-ATTATTTCTAACCTCGACCGC-3′ (reverse primer) and for the unmethylated reaction were 5′-TAGTATAGTATGGTTGGGTTGT-3′ (forward primer) and 5′-ACTATTATTTCTAACCTCAACCAC-3′) (reverse primer). The annealing temperature was 59 °C generating a 106-bp fragment, and the amplified sequence contains 22 CpG sites.

For the *IRS1* gene, the region −1556 to −148 was included (chr2:226797367-226800196). Primers sequences for the methylated reaction were 5′-CGAGGAGATGAAATCGTTATC-3′ (forward primer) and 5′-ATACCGACTTCCCGCTACT-3′ (reverse primer) and for the unmethylated reaction were 5′-ATTTGAGGAGATGAAATTGTTATT-3′ (forward primer) and 5′-CATACCAACTTCCCACTACTC-3′) (reverse primer). The annealing temperature was 61 °C generating a 156-bp fragment, and the amplified sequence contains 244 CpG sites.

The resulting MSP products were extracted from agarose gels using a QIAquick Gel Extraction Kit (Qiagen, Hilden, Germany) and sequenced with ABI BigDye using an ABI 310 sequencer (Applied Biosystems). The sequences obtained were compared with sequences from the NCBI database using BLAST (http://blast.ncbi.nlm.nih.gov/Blast.cgi accessed on 16 November 2020) to confirm amplicon identities.

### 2.7. Statistical Analyses

For the analysis of the parameters of the metabolic control and methylated status of the *INS* and *IRS1* genes, the parameters recommended and not recommended were categorized according to the National Institutes of Health guidelines and the International Diabetes Federation [25,28,29].

#### 2.7.1. Recommended Body Composition Parameters

The recommended body composition parameters were as follows: BMI of 18.5–24.9 kg/m^2^; waist circumference in men of ≤101 cm and in women of ≤87; fat mass in men of ≤24.9% and in women of ≤32.9%; and muscle mass in men of 40–50% and in women of 30–40%. Parameters outside these ranges were considered not recommended.

#### 2.7.2. Recommended Biochemical Parameters

The recommended biochemical parameters were as follows: Fasting glucose of ≤126 mg/dL; triglycerides of ≤149 mg/dL; cholesterol of ≤199 mg/dL; c-LDL of ≤100 mg/dL; uric acid in men of ≤6.9 mg/dL and in women of ≤5.9 mg/dL. Parameters outside these ranges were considered not recommended.

Statistical analyses were performed with GraphPad Prism 8.3.0 software and SPSS version 25. The Mann–Whitney U test was used to compare continuous variables and McNemar’s test was used to compare the categorical dates before and after the intervention with inulin. *p* ≤ 0.05 was considered statistically significant.

### 2.8. Ethical Approval and Informed Consent

The protocol was approved by the General Direction of the Hospital of Huitzuco Guerrero, Mexico and by the Investigation Committee of the State Health Services of the state of Guerrero, Mexico, and was registered in the Book of Protocols of the Department of Health Investigation/SES Guerrero with the folio number 08160718. Each patient signed an informed consent document. 

## 3. Results

The characteristics of the study population before and after the agave inulin intervention are summarized in Table 1. The studied population had a higher prevalence of women (65.7%) with an average age of 54.4 years.

Regarding the anthropometric variables, statistically significant differences were found in weight, BMI, abdominal circumference, and muscle mass after the intervention with agave inulin (Table 1).

We observed a statistically significant decrease in BMI after the intervention with agave inulin ($\overset{-}{x}$ = 30.4 vs. $\overset{-}{x}$ = 29.8; *p* < 0.001). The abdominal circumference was significantly decreased after the intervention ($\overset{-}{x}$ = 99.5) compared to the waist circumference prior to the intervention ($\overset{-}{x}$ = 100.3). Significant changes were also observed in muscle mass values. Regarding the biochemical data, the fasting glucose levels were significantly decreased after the intervention ($\overset{-}{x}$ = 144.7) compared to the initial value ($\overset{-}{x}$ = 164.5), and the triglyceride levels were significantly decreased after the intervention ($\overset{-}{x}$ = 171) compared to the initial value ($\overset{-}{x}$ = 221.3). Both results were slightly elevated in accordance with the ATP III classification. Moreover, cholesterol decreased after the intervention compared to before the intervention, but the difference was not statistically significant.

Next, the methylation, hemimethylation, and unmethylation of the *INS* and *IRS1* genes of the studied population were determined. The *INS* gene was methylated before and after the intervention in 76.1% and 52.2% of the participants, respectively. Concerning the hemimethylated status for the *INS* gene, 23.8% of the studied population presented this status before the intervention with an increase of 23.9% postintervention. For the *IRS1* gene, the hemimethylated status was present in 22.3% and 19.4% of participants pre- and post-intervention, respectively. For the *INS* gene, 77.6% and 80.6% of the participants had an unmethylated status before and after the intervention, respectively (Table 2).

A statistically significant difference was found in the uric acid parameter between the treatment before and after the intervention with agave inulin (Table 3). 

On the other hand, interesting results were observed when we divide the categorized parameters considering the methylation, hemimethylation, and non-methylation of the *INS* and *IRS1* genes before and after the intervention.

As shown in Table 4, the number of people with the methylated *INS* gene decreased in the recommended c-LDL category (*p* = 0.06) and in the recommended uric acid category (*p* = 0.0012) after the intervention with agave inulin. On the other hand, when we found the unmethylated *IRS1* gene, the number of people with non-recommended muscle mass was reduced after the intervention (*p* = 0.01). In contrast, the number of people with recommended uric acid decreased significantly after the intervention with agave inulin when the *IRS1* gene was hemimethylated (*p* = 0.007).

## 4. Discussion

According to the World Health Organization (WHO), excess weight and obesity are major public health problems affecting Mexico, and the characteristics of the population included are consistent with these data [30]. Unfortunately, pre-intervention, we did not find any subjects with normal weight, and most of the population presented uncontrolled biochemical parameters. Post-intervention, four of the nine parameters increased their percentage in the recommended categories. Because metabolic parameters can be modified by several factors, we determined that there were no significant changes in adherence related to the intervention, diet, and physical activity between the beginning and the end of the intervention. Regarding the 13 people who changed from overweight to normal weight, these results reinforce the scientific evidence of the benefits of inulin consumption in humans [23]. However, since the patients were using antidiabetic medications, and these medications have been associated with decreased body weight, these results could be partially related to the intake of glycemic control drugs. 

On the other hand, we found that the number of subjects in the uric acid parameter changed significantly comparing pre-intervention and post-intervention. These results may be associated with the modulation exerted by inulin on the gut microbiome, which is capable of modulating uric acid levels [31,32]. 

Inulin is not digested in the human gastrointestinal system but is fermented by gut bacteria. Through fermentation, the end-products are lactate and short-chain fatty acids, including acetate [33]. We hypothesize that this modulation of the intestinal microbiota explains the control of metabolic parameters due to the decrease in chronic inflammation caused by endotoxins. 

The significant decrease in the methylated levels of the *INS* gene post-intervention with inulin may be due to the induction of DNA demethylation mechanisms. Recently, epigenetic mechanisms involving the intestinal microbiota have taken on great importance in the possible explanation of metabolic diseases such as T2DM. However, few studies have been published addressing this issue. Zhang et al. reported modulation in gene expression associated with epigenetic changes. Previous reports have shown that depending on where the DNA methylation is in the genomic sequence, methylation can have various effects on gene function, such as gene silencing [7]. A previous study on diabetic and nondiabetic participants showed that there are increased levels of methylation at four CpG sites in the promoter region of the *INS* gene in pancreatic islets obtained from diabetic patients compared to nondiabetic patients; the methylation levels of these sites, along with nine other sites in the region, are negatively correlated with the expression of the same gene, which may suggest that the insulin gene is subject to regulation by epigenetic factors [16].

When we divided the recommended and non-recommended parameters of metabolic control including methylation of the *INS* and *IRS1* genes, the BMI classification results were found to be significantly associated with the presence of *INS* gene methylation before and after the intervention in the studied population. Previous epigenome studies have reported that there are CpG islands, approximately 500, which may be methylated in relation to BMI [34,35]. The causality between BMI and DNA methylation has been analyzed, suggesting that BMI and its change have a causal effect on DNA methylation [35].

The nonsignificant change in hemimethylation in the *IRS1* gene may suggest that the gene promoter DNA methylation mechanism may not be affected due to the metabolic changes that occurred in the participants in the agave inulin intervention. It has been reported that epigenetic changes may be the cause of the disease, a consequence of the disease, or a factor that indirectly contributes via environmental exposures that may affect the epigenome [36,37].

In the present study, the results suggested that DNA methylation may be a reversible and dynamic process that requires continuous regulation. Until a few years ago, the mechanisms of a process called demethylation were unknown. According to Bhutani et al., the following two main mechanisms for demethylation are known: Passive demethylation and active demethylation [38]. Passive demethylation is performed by the decrease or absence of DNMT activity, especially DNMT1, during the maintenance of 5mC after cell replication, thus losing methylation in the hemimethylated strand [38]. In contrast, active demethylation occurs in response to changes in cell signals based on two pathways, namely, the oxidative pathway of the methyl group of 5mC and the cytosine deamination pathway induced by cellular repair mechanisms for the generation of a cytosine [38]. However, it is not yet clear how these mechanisms explain the defects found in a disease, such as T2DM, because the active pathways have not yet been characterized under pathological conditions.

Most previous studies that have related methylation levels to blood glucose have reported that increased methylation is associated with prolonged exposure to high glucose levels, which in turn could be associated with a decrease in the expression of the involved genes [11,15,36].

We determined gene methylation in DNA from cells taken from peripheral blood because epigenetic biomarkers in blood have more clinical potential than pancreatic islets as islets cannot be non-invasively assessed. There is evidence that DNA methylation changes in blood cells reflect the epigenetic changes identified in human pancreatic islets and can reflect insulin secretion [39,40,41,42,43]. 

It is also important to mention that since the Glycated hemoglobin (HbA1c) reflects the cumulative glucose exposure of erythrocytes over 3 months and the intervention lasted 2 months, we do not consider it necessary to carry out this measurement [44]; however, since HbA1c is a reliable marker of diabetes control, it would be important to include it in future studies.

There are a few limitations in our study, which could have affected our findings. First, our study population included only 67 participants and may thus be too small to detect differential DNA methylation and strong associations with the characteristics of study participants. Secondly, the MSP technique is qualitative, due to which we were not able to establish quantitative relationships with the metabolic control parameters.

## 5. Conclusions

For the first time, we report the modification in the methylation of two genes, *INS* and *IRS1*, of the insulin pathway and provide information on the possible relevant role of epigenetics as a key factor in positive changes in metabolic control parameters by inulin intake in T2DM patients. 

Further studies are needed to confirm the possible relationship between gene methylation of the insulin pathway and the benefits in metabolic markers of inulin intake. 

## Figures and Tables

**Table 1 nutrients-14-05195-t001:** Characteristics of study population before and after agave inulin intervention.

Parameters	Before	After	*p*
Sex	Men	23 (34.4%)	-
	Women	44 (65.7%)	-
Age (years) X ± ST	54.4 ± 10.2		-
Metabolic control parameters			
Weight (kg)	73.8 ± 13.6	72.5 ± 14.3	0.001
BMI (kg/m^2^)	30.4 ± 04.29	29.8± 04.6	<0.001
Waist circumference (cm)	100.3 ± 10.1	99.5 ± 10.2	0.001
Fat mass (%)	35.2 ± 07.3	35.1 ± 07.6	0.966
Muscle mass (%)	44.5 ± 09.0	43.9 ± 08.9	0.020
Fasting Glucose (mg/dL)	164.5 ± 84.6	144.7 ± 57.6	0.013
Triglycerides (mg/dL)	221.3 ± 196.0	171 ± 101.0	0.008
Cholesterol (mg/dL)	202.7 ± 47.9	194.5 ± 47.9	0.120
c-LDL (mg/dL)	113.6 ± 39.26	115.3 ± 38.9	0.695
Uric acid (mg/dL)	04.9 ± 02.5	04.6 ± 01.4	0.372
Dietary evaluation			
Energy cal	1668 ± 1074	1613 ± 1331	0.792
Proteins	70.42 ± 66.22	60.50 ± 19.92	0.242
Lipids	61.20 ± 68.6	50.14 ± 28.01	0.224
Carbohydrates	209 ± 93.2	194 ± 82.2	0.320
Fiber	19.3 ± 9.60	16.9 ± 8.2	0.129

BMI = body mass index; c-LDL = LDL cholesterol.

**Table 2 nutrients-14-05195-t002:** Methylation status of the *INS* and *IRS1* genes before and after inulin intervention in T2DM patients.

Status	Before		After		
	*n*	%	*n*	%	*p*
*INS*					
Methylated	51	76.1	35	52.2	0.0001
Hemimethylated	16	23.8	32	47.7	
*IRS1*					
Hemimethylated	15	22.3	13	19.4	0.79
Unmethylated	52	77.6	54	80.6	

**Table 3 nutrients-14-05195-t003:** Parameters recommended for metabolic control before and after inulin intervention in T2DM patients.

Parameters	Before		After		*p*
BMI	#	%		%	
Recommended	0	0	13	19.4	
Not recommended	67	100	54	80.5	--
Abdominal circumference					
Recommended	17	25.4	18	26.9	1
Not recommended	50	74.6	49	73.1	
Fat mass					
Recommended	12	17.9	15	22.4	0.37
Not recommended	55	82.1	52	77.6	
Muscle mass					
Recommended	67	100	66	98.5	--
Not recommended	0	0	1	1.5	
Fasting Glucose					
Recommended	30	44.8	35	52.2	0.30
Not recommended	37	55.2	32	47.8	
Triglycerides					
Recommended	32	47.8	32	47.8	1
Not recommended	35	52.2	35	52.2	
Cholesterol					
Recommended	34	50.7	40	59.7	0.28
Not recommended	33	49.3	27	40.3	
c-LDL					
Recommended	24	35.8	20	29.9	0.38
Not recommended	43	64.2	47	70.1	
Uric acid					
Recommended	52	77.6	36	53.7	
Not recommended	15	22.4	31	46.3	0.004

BMI = body mass index; c-LDL = LDL cholesterol.

**Table 4 nutrients-14-05195-t004:** Recommended parameters of metabolic control and methylated status of the *INS* and *IRS1* genes before and after inulin intervention in T2DM patients.

Parameters	Methylated *INS*			Hemimethylated *INS*			Unmethylated *IRS1*			Hemimethylated *IRS1*		
	Before	After	*p*	Before	After	*p*	Before	After	*p*	Before	After	*p*
BMI												
Recommended	0	7		0	6		0	12		0	1	
Not recommended	51	28	-	16	26	-	67	42	-	67	12	-
Waist circumference												
Recommended	15	11		2	7		12	13		5	5	
Not recommended	36	24	1	14	25	0.69	40	41	1	10	8	1
Fat mass												
Recommended	9	9		3	6		7	12		5	3	
Not recommended	42	26	0.37	13	26	1	45	42	0.28	10	10	0.63
Muscle mass												
Recommended	51	34		16	32		52	54		15	12	
Not recommended	0	1	-	0	0	-	52	0	-	15	1	0.01
Fasting Glucose												
Recommended	22	20		8	15		24	27		6	8	
Not recommended	29	15	0.31	8	17	1	28	27	0.60	9	5	0.41
Triglycerides												
Recommended	23	17		9	15		27	26		5	6	
Not recommended	28	18	0.62	7	17	0.76	25	28	0.76	10	7	0.68
Cholesterol												
Recommended	25	20		9	20		27	33		7	7	
Not recommended	26	15	0.48	7	12	0.75	25	21	0.46	8	6	1
c-LDL												
Recommended	21	9		3	11		19	17		5	3	
Not recommended	30	26	0.06	13	21	0.32	33	37	0.71	10	10	0.65
Uric acid												
Recommended	40	15		12	21		41	29		11	7	
Not recommended	11	20	0.0012	4	11	0.74	11	25	0.007	4	6	0.40

BMI = body mass index; c-LDL = LDL cholesterol.

## Data Availability

Supporting Data is available upon request to corresponding author.

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
