# Peer review of "Effect of Inulin Intervention on Metabolic Control and Methylation of *INS* and *IRS1* Genes in Patients with Type 2 Diabetes Mellitus"

_nutrients, 2022, doi:10.3390/nu14235195_

Round 1
Reviewer 1 Report
DNA methylation was determined using a methylation specific PCR. The authors should indicate where the methylation sites in the genes are located. How many unmethylated or methylated cytosine bases are analyzed by the PCR? The PCR product was analyzed to evaluate the presence and saturation of the bands. The authors should explain how the methylation specific PCR has been validated. Did the authors compare the methylation specific PCR with a different method?
The authors have obtained fat mass, abdominal fat and muscle mass from a scale. Are these results reliable?
The tables 4 and 5 show correlations between changes in gene methylation and changes in clinical parameters. However, the gene methylation is shown as a categorical result in table 1. Therfore, the authors should explain the methylation difference which they use for the correlation analysis. Furthermore, the legends of table 4 and 5 give additional correlations (gene methylation level changes and other parameters before or after the intervention?) which do not comply with the correlations shown in the table.
The participants have immediately received a high inulin dose. The usual recommendation is to start the treatment with a smaller dose. How did the participants tolerate this treatment?
The DNA methylation has been investigated in peripheral blood leucocytes which do not produce insulin. The authors should discuss this limitation of the study.
In the abstract the authors do not indicate which changes of the gene methylation were significant and which changes were not significant.
In my opinion the discussion is too long and should be shortened.
Reviewer 2 Report
A very interesting topic and adequately presented. However there are some points need to be clarified
1. English language needs improvement. For example in page 2 the word epigenetics repeated many times, that's why the paragraph needs modification
2. There are some minor spelling errors, for example in discussion line 270 change "in has been" to "it has been"
Authors mentioned that patients received antidiabetic agents. It is well known that some of these agents cause weight loss. Did they took it into consideration?
2 months is a very sort period in order to exclude safe results. In my opinion authors should mention the limitations of their study
Methods in abstract should be more detailed
They do not mentioned if their results differ according to HbA1c levels
In study design they mention that patients were enrolled after invitation by telephone. I think this is not acceptable, because some of the participants might have communication problems
Round 2
Reviewer 1 Report
Minor comments:
line 212: waist circumference men ≤ 101 cm
line 289: Next
Author Response
We appreciate your comments.
We have corrected the errors in the text
Reviewer 2 Report
Authors made grate effort and improved their manuscript. However, as long as HbA1c is a marker showing the diabetes control they should also mention that they did not included it in their study. In addittion, even if previous studies lasted for the same period, i think that if they want to prove that these substances improve glycemic the control, the positive affection should be the improvement of HbA1c which is usually a reliable marker. As long as their study lasted only two months they could not calculate it
It is important to mention if they had difficulties in patients enrollment how they encountered these difficulties
Although authors answered that weight loss is not due to medication, they should mention that it could be because of this and if they believe it is not, they should explain why they believe it. For example GLP-1 analogues and sglt2 inhibitors cause weight loss. How can they be sure that these substances are the cause and not these medications?
